# FLIQS: One-Shot Mixed-Precision Floating-Point and Integer Quantization Search

Jordan Dotzel[1,2]  Gang Wu[2]  Andrew Li[2]  Muhammad Umar[1]
Yun Ni[2]  Mohamed S. Abdelfattah[1]  Zhiru Zhang[1]
Liqun Cheng[2]  Martin Dixon[2]  Norman P. Jouppi[2]  Quoc V. Le[2]  Sheng Li[2]

[1]Cornell University
[2]Google

**Abstract**  Quantization has become a mainstream compression technique for reducing model size, computational requirements, and energy consumption for modern deep neural networks (DNNs). With improved numerical support in recent hardware, including multiple variants of integer and floating point, mixed-precision quantization has become necessary to achieve high-quality results with low model cost. Prior mixed-precision methods have performed either a post-training quantization search, which compromises on accuracy, or a differentiable quantization search, which leads to high memory usage from branching. Therefore, we propose the first one-shot mixed-precision quantization search that eliminates the need for retraining in both integer and low-precision floating point models. We evaluate our search (FLIQS) on multiple convolutional and vision transformer networks to discover Pareto-optimal models. Our approach improves upon uniform precision, manual mixed-precision, and recent integer quantization search methods. With integer models, we increase the accuracy of ResNet-18 on ImageNet by 1.31% points and ResNet-50 by 0.90% points with equivalent model cost over previous methods. Additionally, for the first time, we explore a novel mixed-precision floating-point search and improve MobileNetV2 by up to 0.98% points compared to prior state-of-the-art FP8 models. Finally, we extend FLIQS to simultaneously search a joint quantization and neural architecture space and improve the ImageNet accuracy by 2.69% points with similar model cost on a MobileNetV2 search space.

## 1 Introduction

In recent years, deep neural networks (DNNs) have achieved state-of-the-art results on a wide range of tasks including image classification, speech recognition, image and speech generation, and recommendation systems. Each model iteration typically enhances quality but also tends to increase computation, memory usage, and power consumption. These increases limit DNN adoption in resource-constrained edge devices, worsen their latency across platforms, and expand their carbon footprint, especially within cloud systems. DNN quantization to low-precision formats has become the standard method for reducing model storage size, memory bandwidth, and complexity of MAC operations [1, 2]. These formats include both integer and low-precision floating-point, which has recently gained attention as a flexible alternative to integer formats.

At the same time, DNN accelerators have become more diverse and now support a wide range of numerical formats. For example, the Google TPUv3 supports FP32, BF16, FP16, and INT8 [3], while the latest NVIDIA Hopper architecture supports FP32, BF16, FP8, and INT8 [4]. Furthermore, reprogrammable systems such as FPGA devices allow arbitrary precision arithmetic such as INT5, FP11, FP9, or FP8 for more granular accuracy-performance trade-offs [5]. While these devices enable mixed-precision quantization, where layers take on different formats within the same model, it is challenging to optimally assign per-layer formats since layers exhibit different quantization characteristics. In simple cases, this assignment can be performed manually, yet with the explosion

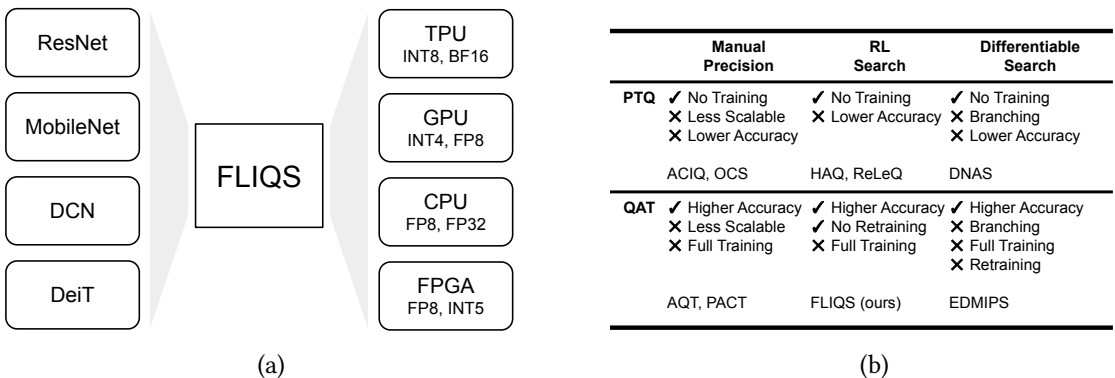

|  | Manual Precision | RL Search | Differentiable Search |
|---|---|---|---|
| **PTQ** | ✓ No Training
✗ Less Scalable
✗ Lower Accuracy | ✓ No Training
✗ Lower Accuracy | ✓ No Training
✗ Branching
✗ Lower Accuracy |
| | ACIQ, OCS | HAQ, ReLeQ | DNAS |
| **QAT** | ✓ Higher Accuracy
✗ Less Scalable
✗ Full Training | ✓ Higher Accuracy
✓ No Retraining
✗ Full Training | ✓ Higher Accuracy
✗ Branching
✗ Full Training
✗ Retraining |
| | AQT, PACT | FLIQS (ours) | EDMIPS |

(a)                                          (b)

Figure 1: **FLIQS** – The explosion of model architectures, numerical support, and deployment platforms requires automated methods for searching model configurations to utilize platform-specific numerical formats. We establish FLIQS as the first one-shot quantization and neural architecture search framework for searching for integer and floating point formats.

of DNN architectures and accelerator designs, automated methods are more reliable, scalable, and reproducible for achieving high accuracy and performance.

In this paper, we introduce F̲Loating-Point and I̲nteger Q̲uantization S̲earch (FLIQS) to automate mixed-precision floating-point and integer quantization and automatically assign per-layer formats. In addition, FLIQS can jointly optimize for quantization formats and neural architecture to intelligently allocate compute across the kernel, channel, and bitwidth dimensions. FLIQS is a one-shot search based on reinforcement learning (RL) and unlike expensive multi-trial searches, it avoids training separate models for each configuration, leading to overall reduced search overhead. Furthermore, as the search takes place during training, FLIQS can achieve higher accuracies than post-training quantization (PTQ) searches. Coupled with additional entropy regularization, the final model can be deployed without the need for further retraining or fine-tuning. As shown in Figure 1(a), FLIQS accelerates the process of adapting legacy models to new hardware, co-designing models and accelerators, and finding Pareto-optimal models on current hardware systems. We summarize our contributions as follows:

1. Introduce the first one-shot quantization search without retraining through the addition of a new cosine entropy regularization schedule;

2. Demonstrate state-of-the-art results for integer and low-precision floating-point quantization search across a range of convolutional and transformer networks;

3. Perform the largest comparison of integer and floating-point mixed-precision networks;

4. Conduct the first study of quantization and neural architecture search on low-precision floating-point networks and establish recommendations for allocating compute across bitwidth and neural architectural dimensions.

## 2 Related Work

**Low-Precision Floating Point**: Low-precision floating point is being discussed as the next generation format for DNN training and inference. [6]. Companies, including AMD, Intel, NVIDIA, and Qualcomm, have recently agreed to adopt 8-bit floating-point (FP8) in future deep learning systems. Within these formats, recent studies generally focus on two variants: E5M2 and E4M3, where E represents the number of exponent bits and M is the number of mantissa bits. For example, HFP8 suggests using E4M3 for the forward pass and E5M2 for backpropagation [7]. Building upon these uniform precision works [7, 8, 9, 10, 11], FLIQS proposes an automated approach for finding

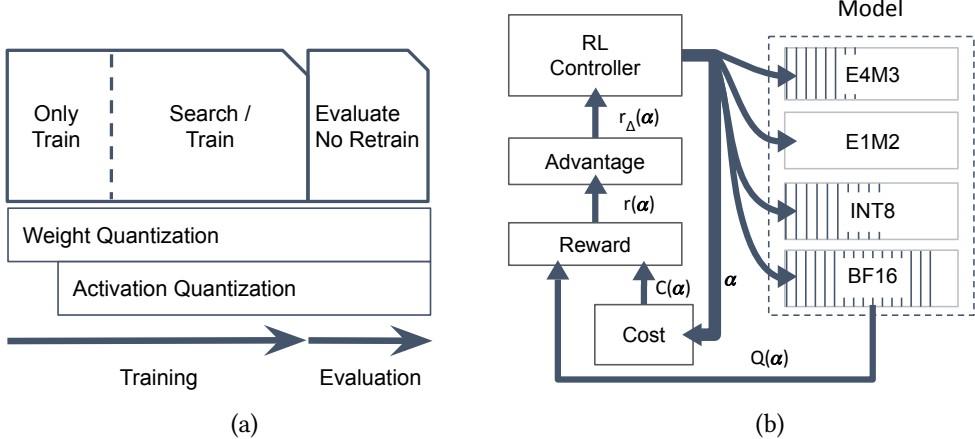

Figure 2: **FLIQS Overview** – (a) FLIQS begins with pure training to allow the reward signal to stabilize before updating its policy. The activation quantization is delayed to allow the activation statistics to stabilize. (b) The RL then controller proposes per-layer formats and architectural decisions during training

mixed-precision floating-point networks, compares these to mixed-precision integer networks with similar cost, and performs a joint floating-point quantization and neural architecture search.

**Quantization Search**: Prior work has explored mixed-precision integer quantization searches, as shown in Figure 1(b). For instance, HAQ [12] and ReLeQ [13] both perform PTQ quantization searches that utilize RL to allocate bitwidths based on the model accuracy and cost estimates. In addition, the HAWQ series of works further develops these PTQ searches, using the Hessian spectrum to determine layer sensitivities and constrained ILP formulations to find optimal bitwidth configurations [14, 15, 16]. However, being PTQ-based, these methods cannot take advantage of the higher accuracy and more accurate feedback provided by quantization-aware training (QAT) during the search.

Other efforts perform quantization search during training, often using neural architecture search (NAS) with super-networks or differentiable NAS [17, 18, 19, 13, 20]. For instance, MPQ uses an adaptive one-shot method that trains models using multiple bitwidths and automatically freezes the bitwidths of specific layers during training to improve the model convergence across bitwidths [21]. In addition, EDMIPS creates branches for each bitwidth, forms a linear combination of them, and then alternates training the layer weights and the branch weights [22]. These differentiable searches often have simpler formulations since the layer and branch weights are unified and trained together with gradient descent. However, because they replicate the weights and activations, they incur higher memory and computational costs compared to RL-based methods. In addition, both PTQ and QAT prior works require additional retraining steps on the model after the search, while FLIQS directly serves the final model without fine-tuning.

**Quantization Neural Architecture Search (QNAS)**: In addition, prior work has explored joint search spaces with quantization formats and neural architecture [23, 24, 25, 26, 27]. For example, APQ uses knowledge distillation from a full-precision accuracy predictor to optimize neural architecture, quantization formats, and pruning policies [25]. FLIQS expands on this line of work by jointly searching quantization formats and neural architecture and highlights trends for allocating compute across this joint search space for high accuracy and performance.

## 3 FLIQS Framework

As a one-shot method, FLIQS employs a controller to sample per-layer formats and model architectures during training. This method allows the search and model to adapt to each other yet it comes with certain challenges. First, the search may interfere with the original model training

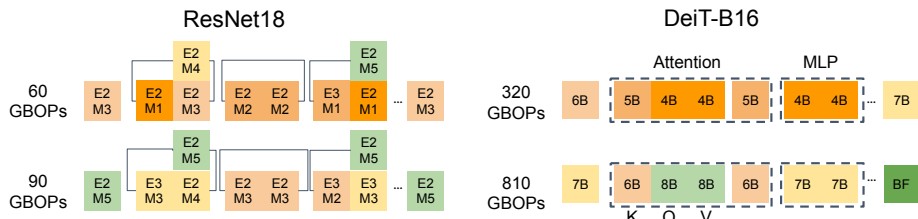

Figure 3: **FLIQS Examples** – In these quantization search examples, FLIQS allocates more precision to the first and last layers and the small pointwise convolutions of ResNet-18, and to the attention block within DeiT-B16. More configurations are listed in Appendix A.1.

process, since modifying the architecture shifts the weight and activation distributions during training. In addition, one-shot search needs to evaluate the quality signal of different architectures on different batches of training data to avoid lengthening the training process. This introduces noise into the reward signal since different batches may have significantly different quality. Also, the controller and policy model must be efficient enough to be embedded within the training graph to not significantly increase the training time. This section addresses these challenges, while focusing on the search space involving per-layer formats and channel widths.

As shown in Figure 2, the model first trains without search, and the architecture is sampled uniformly at random to avoid overfitting to a single option. It uses standard fake quantization and employs a two-phase approach that delays activation quantization to improve stability (Appendix A.2). Next, at each training step, the controller proposes a new architecture and applies it to the model. The model then performs a standard forward and backward pass on the training data to produce the model gradients and a forward pass on the validation data to produce a quality signal for the controller. This quality signal is combined with the model cost in Figure 2(b) to produce a reward and reward advantage, which the controller then uses to update its policy. After the search and training finish, the model is directly used for inference without additional fine-tuning or retraining.

**Cost and Reward Function**: FLIQS uses the quadratic cost model, bit operations (BOPs), as described in Equation 1 where $b(\alpha)$ is the total bitwidth of the current layer architecture $\alpha$ and $MAC_l(\alpha)$ represents the number of multiply-accumulates (MACs) in layer $l$. Quadratic cost models, which predict power and area, are particularly useful in model-accelerator co-design where multipliers dominate resources and scale quadratically in power and area [28].

$$C_l(\alpha) = b(\alpha)^2 \cdot MAC_l(\alpha), \quad r(\boldsymbol{\alpha}) = Q(\boldsymbol{\alpha}) + \gamma \left| \frac{\sum_l C_l(\alpha)}{C_T} - 1 \right| \tag{1}$$

This model cost is combined with the quality signal, $Q(\alpha)$, in the absolute reward function shown in Equation 1 [29]. This quality signal is model and application dependent but in the simple case is the validation accuracy. The absolute reward function includes a cost target $C_T$ that provides the user control over the accuracy-performance trade off. More restrictive targets tend to result in less compute-intensive models (as shown in Figure 3), which often have lower accuracy. This resultant cost term is combined with the model quality using the cost scalar $\gamma$, which balances the importance of performance and quality.

**RL Controller**: The RL controller is in charge of choosing the model architecture at each step. It learns a policy $\pi_l(\alpha)$ for each layer $l$ that represents a probability distribution over each architecture $\alpha$. At each training step, the controller samples and applies a new layer architecture $\alpha_l \sim \pi_l(\alpha)$. The channel widths are efficiently searched by applying channel masks, which dynamically zero out channels and reuse the underlying weights during training. This policy $\pi_l(\alpha)$ is parameterized by $\theta_{l,\alpha}$, where $\theta_{l,\alpha}$ represents the logit for the $\alpha^{th}$ decision in the $l^{th}$ layer. These logits are then passed through a softmax layer to produce the policy probability distribution.

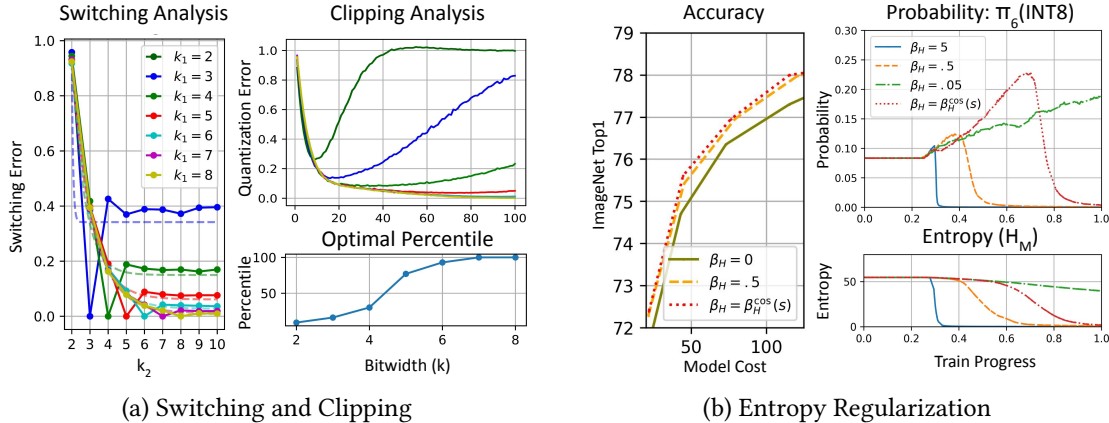

(a) Switching and Clipping          (b) Entropy Regularization

Figure 4: **FLIQS Analysis** – (a) The switching error grows relatively large when either bitwidth is small and affects model convergence. In addition, the optimal clipping threshold depends on the current bitwidth, which motivates swapping thresholds. (b) Accuracy improves for higher entropy regularization, and the entropy regularization affects the policy convergence.

$$\pi_l(\alpha) = \frac{\exp(\theta_{l,\alpha})}{\sum_j \exp(\theta_{l,j})} \tag{2}$$

After sampling and assigning the model architecture, $\boldsymbol{\alpha}$, the reward $r(\boldsymbol{\alpha})$ is calculated according to Equation 1. However, since the reward depends on the quality signal, which increases throughout training, the difference between the running average of previous rewards, $\bar{r}(\boldsymbol{\alpha})$, and the current reward is used instead: $r_\Delta(\boldsymbol{\alpha}) = \bar{r}(\boldsymbol{\alpha}) - r(\boldsymbol{\alpha})$. Then, the REINFORCE algorithm [30] is used to update the policy $\pi_l(\alpha)$ by performing gradient descent on the policy loss, $\mathcal{L}_\theta$:

$$\mathcal{L}_\theta = -r_\Delta(\boldsymbol{\alpha}) \sum_l \log\left(\alpha_l \sim \pi_l(\alpha)\right), \quad \theta \leftarrow \theta + \eta \nabla_\theta \mathcal{L}_\theta \tag{3}$$

where $\eta$ is the RL learning rate. This procedure is chosen due to its low complexity, and it helps address the performance concerns with one-shot searches (analysis shown in Appendix A.6). Other reinforcement learning methods, such as PPO, and more sophisticated policy models, such as multi-layer perceptron models, offered no quality improvements while being more costly.

**Format Search Space**: For pure quantization search, this work evaluates FLIQS on two search spaces: FLIQS-S and FLIQS-L. FLIQS-S includes the standard power-of-two formats, while FLIQS-L includes a larger set of formats between four and eight bits. For floating point, FLIQS-L includes 16 formats, which to our knowledge is the largest quantization search performed. Full details of the quantization search spaces can be found in Appendix A.5.

**Switchable Clipping**: FLIQS also introduces a switchable clipping threshold that changes based on the current format. This is necessary since smaller bitwidths require more aggressive clipping, and vice versa, as shown in Figure 4(a). These clipping thresholds can either be pre-computed with synthetic data, or computed during the first phase of the search with real data. In general, pre-computing the thresholds leads to high-quality results with less complexity, and it is used for the experimental sections below.

## 4 FLIQS Analysis

**Switching Error**: The primary challenge for FLIQS is minimizing the effect of the search on the model training. Within a pure quantization search, this effect can be formalized by introducing the *switching error*. Consider the standard symmetric integer quantizer, $Q(x; s)$ with the scale factor

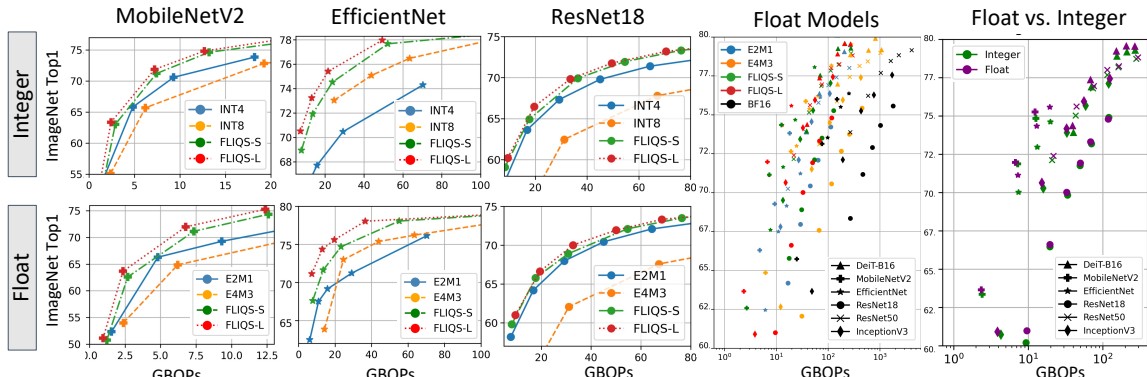

Figure 5: **ImageNet FLIQS Quantization Search** – FLIQS reaches higher accuracies at lower costs, and in general FLIQS-L achieves higher accuracies. Models are evaluated at multiple widths ranging .25× to 2× of their original channel width to generate each data point.

$s = (2^{k-1} - 1)/\sigma_T$, where $\sigma_T$ is the clipping threshold. This gives the absolute quantization error $\Delta(x; s)$, defined as:

$$Q(x; s) = \lfloor x \cdot s \rceil /s, \quad \Delta(x; s) = |Q(x; s) - x| \tag{4}$$

For a fixed $\sigma_T$, $Q(x; s)$ and $\Delta(x; s)$ can instead be parameterized solely by the bitwidth $k$. When this varies during the search, it produces a switching error:

$$\Delta_S(x; k_1, k_2) = |Q(x; k_2) - Q(x; k_1)| \tag{5}$$

As illustrated in Figure 4(a), this switching error for standard search spaces, such as integer FLIQS-S, can be relatively large (setup details listed in Appendix A.8).

**Convergence**: This switching error can be viewed as an additional source of noise for the model optimizer, typically SGD or Adam [31]. Intuitively, the expected switching error should be proportional to the total policy entropy $H_M$ of the model $M$:

$$H_M = - \sum_{l \in M} \sum_k \pi_l(k) \log \pi_l(k), \quad E[\Delta_S(x; k_1, k_2)] \propto H(\pi_l) \tag{6}$$

That is, as the policy decreases entropy over time by settling on specific formats, the expected switching error decreases and converges to zero as the entropy tends toward negative infinity. This can be seen explicitly by modeling $\pi_l(k) \sim N(k; \mu, \sigma)$ as a Gaussian distribution, which has an entropy $H = \frac{1}{2} \log(2\pi e \sigma^2)$. Under these assumptions, $\lim_{H \to -\infty} \Rightarrow \lim_{\sigma \to 0} \Rightarrow \lim_{k_1 \to k_2}$ and thus:

$$\lim_{H \to -\infty} E[\Delta_S(x; k_1, k_2)] = E[\lim_{k_1 \to k_2} \Delta_S(x; k_1, k_2)] = E[\Delta_S(x; k_2, k_2)] = 0 \tag{7}$$

since $\Delta_S(x; k, k) = 0$. Therefore, as the model entropy decreases, the search no longer interferes with the model training, and this interference can be formulated in terms of additional optimization noise. The noise ball around the optimum is proportional to the entropy, and therefore convergence requires carefully controlling the entropy.

**Entropy Regularization**: FLIQS introduces entropy regularization to reduce the entropy toward the end of the search and enable searches without a final retraining. This addresses the key challenge of one-shot quantization search by diminishing the effects of the search on the model training. The entropy regularization adds a new loss term to the policy loss $\mathcal{L}_\theta$, balanced by a factor $\beta_H$.

$$\mathcal{L} = \mathcal{L}_\theta - \beta_H H_M \tag{8}$$

$$\beta_H^{\cos} = -.5\beta_H^{end}(1 + \cos(\pi s)) + \beta_H^{end} \tag{9}$$

Table 1: **Quantization Search** – 'GBOPS' is the model cost given in billions of bit-ops, and '*' indicates the first and last layers are kept in higher precision. The mean and standard deviations are listed for FLIQS methods, aggregated over three trials.

| Method | Precision | ResNet-18 | | ResNet-50 | | MobileNetV2 | |
|---|---|---|---|---|---|---|---|
| | | GBOPs | Top-1 | GBOPs | Top-1 | GBOPs | Top-1 |
| BF16 | 16 | 467 | $72.80_{0.16}$ | 1047 | $78.05_{0.05}$ | 77 | $73.13_{0.14}$ |
| HAWQ-V3 [16] | 4* | 34 | 68.45 | 71 | 74.24 | - | - |
| ZeroQ [32] | 2,8 | - | - | 70 | 76.08 | 5 | 69.44 |
| EDMIPS [22] | [1,4] | 22 | 67.20 | 49 | 73.20 | - | - |
| LQNets [33] | 4* | 34 | 69.30 | 71 | 75.10 | - | - |
| INT FLIQS-S | 4,8,16 | $31_{0.06}$ | $69.91_{0.18}$ | $73_{1.43}$ | $77.40_{0.12}$ | $7_{0.03}$ | $71.21_{0.18}$ |
| INT FLIQS-L | [4,8],16 | $32_{0.17}$ | $70.61_{0.04}$ | $72_{0.53}$ | $77.31_{0.03}$ | $7_{0.09}$ | $71.87_{0.24}$ |
| HAWQ-V3 [16] | 4, 8* | 72 | 70.38 | 154 | 76.73 | - | - |
| Bayesian Bits [34] | [2,32] | 56 | 69.80 | - | - | 17 | 72.00 |
| DQ [35] | [2,10] | 226 | 70.08 | - | - | 37 | 69.74 |
| PACT [36] | 5* | 50 | 69.80 | 101 | 76.70 | - | - |
| INT FLIQS-S | 4,8,16 | $48_{1.61}$ | $71.23_{0.10}$ | $81_{1.25}$ | $77.32_{0.05}$ | $17_{0.73}$ | $72.98_{0.22}$ |
| INT FLIQS-L | [4,8],16 | $43_{1.10}$ | $71.51_{0.10}$ | $80_{2.30}$ | $77.34_{0.05}$ | $17_{0.06}$ | $72.96_{0.26}$ |
| HFP8 [7] | 8* | 137 | 69.39 | 284 | 76.22 | 21 | 71.61 |
| FPQuant [9] | 8 | 116 | 70.28 | - | - | 19 | 71.60 |
| MPFP [8] | 8* | 137 | 69.71 | 284 | 75.70 | - | - |
| FP FLIQS-L | [4,8],16 | $46_{1.01}$ | $71.64_{0.37}$ | $74_{0.51}$ | $77.34_{0.14}$ | $17_{0.32}$ | $72.94_{0.09}$ |

In addition, FLIQS introduces a cosine entropy regularization schedule in Equation 9, where $s \in [0, 1]$ represents the current training progress and $\beta_H^{end} = 0.5$. Figure 4(b) demonstrates the characteristics of this schedule and the tradeoffs in choosing $\beta_H$. It can achieve high quality results through high exploration at the beginning of the search (high $H_M$) and final stability for the quantization-aware training at the end. Appendix A.11 demonstrates that retraining after the search adds no benefit with entropy regularization.

## 5 Quantization Search

We begin by evaluating FLIQS on pure quantization search spaces, since this allows comparisons to the most previous work. All models were trained from scratch with cloud-based TPUv3 cluster, and all training and search hyper-parameters are listed in Appendix A.3.

**Pareto Curves**: Figure 5 shows the Pareto curves for uniform precision and FLIQS models. It demonstrates that FLIQS outperforms uniform precision methods across ImageNet models, often with large margins. The FLIQS-L searched models lead to the highest accuracy overall, yet this search space requires support for arbitrary precision in hardware, e.g. within FPGA platforms. In addition, when comparing models together, the FLIQS-L MobileNetV2 outperforms all others models across floating-point and integer formats, with FLIQS-L EfficientNet following closely behind. Finally, the integer and floating-point models are plotted together and show that in nearly every case, floating-point outperforms integer.

To achieve these results, FLIQS makes different decisions for each model guided by the reward signal. For the ResNet models, it assigns most layers to low-precision, except for the first and last. It further increases the precision of the pointwise convolutions in the downsampling skip branches (the top 8B convolutions in Figure 3). In contrast, for EfficientNet and MobileNetV2 the pointwise convolutions are typically in lower precision while the depthwise convolutions are in higher precision. Lastly, the vision transformer model, DeiT, shows similar behavior to the other

**ResNet Performance** – The ResNet18 area estimates demonstrate a small impact from the additional layers in higher precision with FLIQS-L and additionally show the correlation between GBOPs and area. The precision column for each of the three layers in the ResNet-18 downsampling block: 3×3, 3×3, 1×1. The ResNet50 results demonstrate that the integer FLIQS-S mixed-precision model does not add significant overhead over HAWQ-V3. FPGA results were gathered on the Xilinx UltraScale+ FPGA platform, where look-up tables (LUTs) are the primary resource.

| Method | Prec. | LUTs | Rel. × | GBOPs | Top-1 |
|---|---|---|---|---|---|
| 4B | 4,4,4 | 42.8K | 1.00× | 29 | $67.31_{0.10}$ |
| 5B | 5,5,5 | 44.8K | 1.05× | 45 | $68.56_{0.13}$ |
| 6B | 6,6,6 | 48.3K | 1.13× | 65 | $69.03_{0.09}$ |
| 7B | 7,7,7 | 54.9K | 1.28× | 89 | $70.32_{0.07}$ |
| 8B | 8,8,8 | 67.6K | 1.58× | 117 | $70.78_{0.10}$ |
| FLIQS-L | 5,5,6 | 45.9K | 1.07× | 46 | $70.12_{0.07}$ |
| FLIQS-L | 5,6,6 | 47.1K | 1.10× | 67 | $71.51_{0.10}$ |

Table 2: ResNet18 Estimated Area

| Method | GBOPs | Speedup (×) | | Top1 |
|---|---|---|---|---|
| | | 2080 Ti | A6000 | |
| INT8 | 262 | 1.000 | 1.000 | $77.47_{0.09}$ |
| INT4 | 65 | 1.338 | 1.234 | $74.91_{0.15}$ |
| INT4* | 71 | 1.334 | 1.228 | $76.31_{0.15}$ |
| FLIQS-S | 73 | 1.303 | 1.213 | $77.40_{0.12}$ |

Table 3: ResNet50 GPU Latency

models in terms of its first and last layers and also allocates more bits to its self-attention blocks. All of the detailed configurations can be found in Appendix A.1.

**Table Comparison**: Table 1 further evaluates FLIQS against previous work. As shown in this table, FLIQS improves overall accuracy while simultaneously reducing the model cost in most cases. For example, it outperforms the recent mixed-precision QS method HAWQ-V3 [16] across multiple model cost targets. For ResNet-50, FLIQS improves the Top-1 accuracy by 0.61% while using only 51% of its GBOPs. In addition, FLIQS-L outperforms many recent works on FP8 model inference. For example, against MPFP [8] on ResNet18, FLIQS finds a variant with 1.93% higher accuracy with a third of the model cost by allocating more bits to the downsampling convolutions and first convolutions in the network.

These results demonstrate that the searched models consistently outperform their uniform precision baselines. Moreover, this section to our knowledge shows the first large-scale comparison of floating-point and integer mixed-precision models and shows that floating-point models outperform their integer counterparts for the same total bitwidth. Joint integer and floating-point searches were attempted; however, since floating-point dominates integer formats at the same total bitwidths, the outputs of these searches were the same as the pure floating-point searches.

**Performance**: To evaluate the performance of the searched models, we use an infrastructure developed by the authors of HAWQV3 [16] that extends the TVM [37] compiler to support INT4 inference. Table 3 shows that on Turing GPUs, the FLIQS-S model improves accuracy significantly with only 1% lower inference speed compared to the INT4 model. In addition, Table 2 shows that LUTs scale quadratically with the precision bitwidth, and since LUTs act as a proxy for area, this verifies the usefulness of the BOPs cost model. This table also confirms the overhead from these searched models is relatively small compared to the accuracy improvements shown in Table 1.

## 6 Quantization Neural Architecture Search

FLIQS can efficiently traverse large quantization search spaces and achieve Pareto-optimal combinations of accuracy and model cost within fixed model architectures. Yet, further improvements can come from combining the quantization search of FLIQS with neural architecture search, which is referred to as FLIQNAS in this section.

Figure 6 evaluates this method on a MobileNetV2 search space, which incorporates tunable filter widths on inverted bottleneck projection layers and adjustable kernel sizes on central depthwise

| Method | Precision | GBOPs | Top1 |
|---|---|---|---|
| MobileNetV2 | 8 | 19 | $72.83_{0.24}$ |
| FLIQNAS-S | 4,8,16 | $13_{0.34}$ | $73.79_{0.14}$ |
| FLIQNAS-L | [4,8],16 | $13_{0.25}$ | $74.79_{0.08}$ |
| APQ-A | 2,4,6 | 13 | 72.10 |
| FLIQS-S | 4,8,16 | $17_{0.73}$ | $72.98_{0.22}$ |
| FLIQS-L | [4,8],16 | $17_{0.21}$ | $72.96_{0.26}$ |
| FLIQNAS-S | 4,8,16 | $17_{0.27}$ | $75.17_{0.08}$ |
| FLIQNAS-L | [4,8],16 | $17_{0.14}$ | $75.65_{0.20}$ |
| APQ-B | 2,4,6 | 16 | 74.10 |
| FLIQNAS-S | 4,8,16 | $21_{0.21}$ | $75.71_{0.11}$ |
| FLIQNAS-L | [4,8],16 | $22_{0.29}$ | $75.95_{0.04}$ |
| APQ-C | 2,4,6 | 23 | 75.10 |

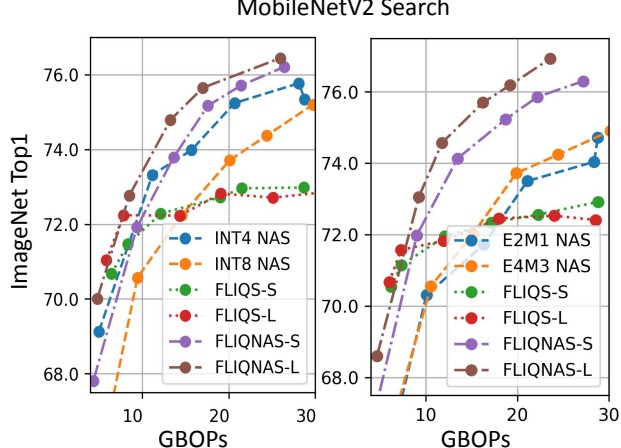

Figure 6: **MobileNetV2 FLIQNAS** – FLIQNAS outperforms APQ in similar search spaces. In addition, the combination of quantization search and neural architecture search outperforms the two methods separately on integer and floating-point formats.

layers. Altogether, there are 230 tunable values leading to a search space of over $10^{100}$ configurations for FLIQNAS-S. This search space is significantly larger than that of the original MobileNetV2 FLIQS-S with 53 options and approximately $10^{25}$ configurations.

This figure compares FLIQNAS to FLIQS and quantized NAS, which fixes the quantization format for all layers and only searches for the architecture. It shows that FLIQS-S and FLIQS-L searches perform well for low model costs, yet as the model scales to higher costs, the compute is better allocated by increasing the size of the architectural components. In this region, both quantized NAS and FLIQNAS yield the best performance. For all model costs, FLIQNAS-L is able to reach the Pareto-optimal tradeoff of accuracy and model cost. Lastly, when compared at identical cost targets, floating-point FLIQNAS surpasses the performance of the integer search space.

In Figure 6, we include a FLIQNAS comparison against APQ [25], which performs a joint architecture, pruning, and quantization search by using a large once-for-all network. Its search space is similar and includes multiple kernel sizes, channel widths, and integer bitwidths built on top of the original MobileNetV2 architecture. This table shows that for similar GBOPs, FLIQNAS leads to higher accuracy over APQ across its three published design points. Further layer-wise analysis of these results is located in Appendix A.7.

# 7 Conclusion

As AI hardware supports an increasing number of numerical formats, DNN quantization search to integer and low-precision floating-point grows increasingly important for reducing memory and compute. This paper proposes FLIQS, the first one-shot RL-based integer and low-precision floating-point quantization search without retraining. Compared to prior work, FLIQS can achieve higher accuracy without involving additional fine-tuning or retraining steps by introducing a cosine entropy regularization schedule. Moreover, as an RL-based method, it reduces the amount of memory needed for weights, activations, and gradients during the search compared to recent differentiable NAS searches.

These enhancements accelerate research progress and enable quantization searches on larger search spaces and more substantial models, such as DeiT-B16, which has 10 times the model cost as BF16 MobileNetV2. In addition, FLIQS conducts the first floating-point quantization search and produces mixed-precision models that outperform the latest works on FP8 formats. When further combined with architecture search, it identifies even stronger MobileNetV2 models than NAS and quantization search alone. It further suggests that for a fixed compute budget, larger models benefit

from increasing architectural dimensions over bitwidth. Overall, FLIQS represents an efficient framework for searching multi-precision models on current hardware and gives further insight into model and hardware co-design for future accelerator generations.

## 8 Acknowledgements

We also acknowledge Lukasz Lew and Daiyi Peng for their invaluable contributions to the project through AutoML and quantization frameworks and multiple discussions. In addition, we thank Andrew Butt for his discussions

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

## A  Appendix

The following sections contain additional experimental details, small experiments, ablation studies, and example output bitwidths. The listed hyper-parameters attempt to make the results more reproducible and interpretable. In addition, the small-scale experiments motivate certain hyper-parameter selections discussed in the main paper. And finally, the example configurations give more insight into how FLIQS allocates bitwidth across different models and cost targets.

### A.1  Example Configurations

FLIQS bitwidth configurations vary based on the model and search space. Figure 7 shows a set of configurations for FLIQS-L and FLIQS-S searches on a ResNet18 across four different model cost targets. Lower bitwidths are represented with colors closer to red and higher bitwidths are closer to green. This figure shows that FLIQS typically gives higher bitwidth to the first and last layers of the model. It also consistently gives higher bitwidths to the 1x1 convolution on the upper branch, and although not obvious in this figure, it usually allocates more bitwidth to the earlier stages of the model compared to later stages.

Figure 8 shows example bitwidth configurations for all models evaluated. It reveals that ResNet50 has similar trends to ResNet18: more bitwidth for the first and last layers, 1x1 convolutions on the upper branch, and generally more in the early stages. Unlike the ResNet models, MobileNetV2 has a main block that comprises a sequence of a pointwise convolution, depthwise convolution, and then pointwise convolution. FLIQS allocates more bitwidth to the central 3x3 depthwise convolution in this block (groups of three in the figure). InceptionV3 has a more complicated branched architecture of 1x1, 3x3, and 5x5 convolutions. This block is shown in the figure as the repeated structure of one, three, two, and then one convolution, from top to bottom. FLIQS likewise gives more bitwidth to the earlier stages of InceptionV3 and its first and last layers. Additionally, it increases the precision of the 1x1 convolutions on the top and bottom of the repeated block.

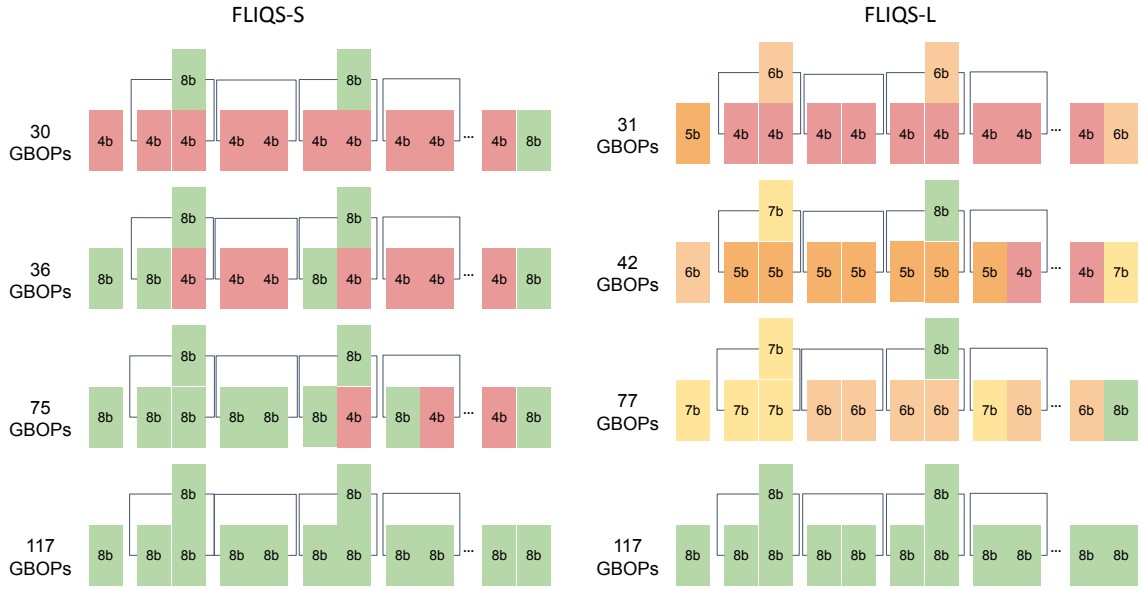

Figure 7: **ResNet18 Integer FLIQS** – Output configurations depend on the model, model cost target, and supported bitwidths. FLIQS-S uses 4 and 8 bits as the search space, while FLIQS-L uses 4 to 8 bits, inclusive. For both variants, FLIQS generally allocates higher bits to the first and last layers, with a slight preference for the last layer. It also assigns more bits to the small upper 1x1 convolutions and more bits to the first 3x3 convolution within a block.

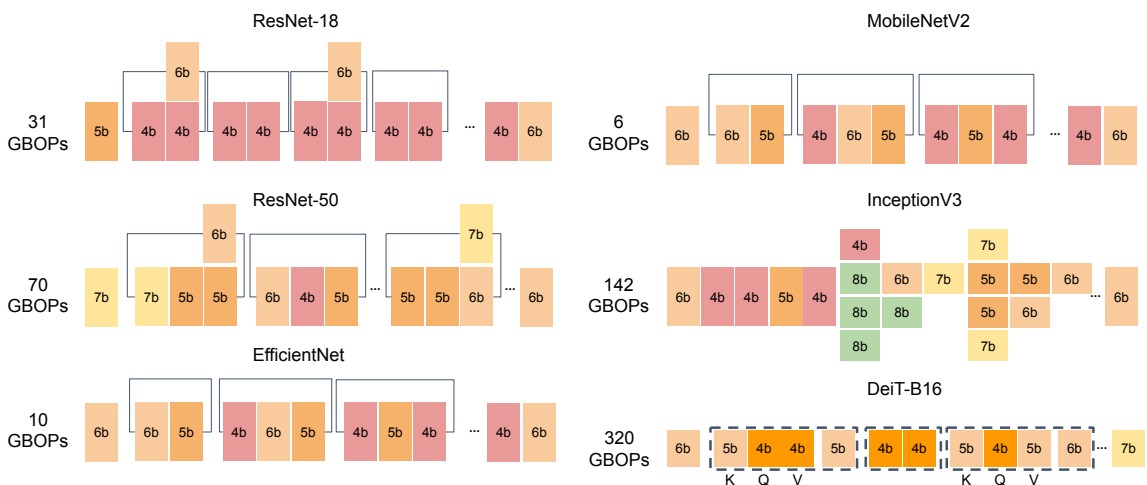

Figure 8: **Integer FLIQS-L Examples**

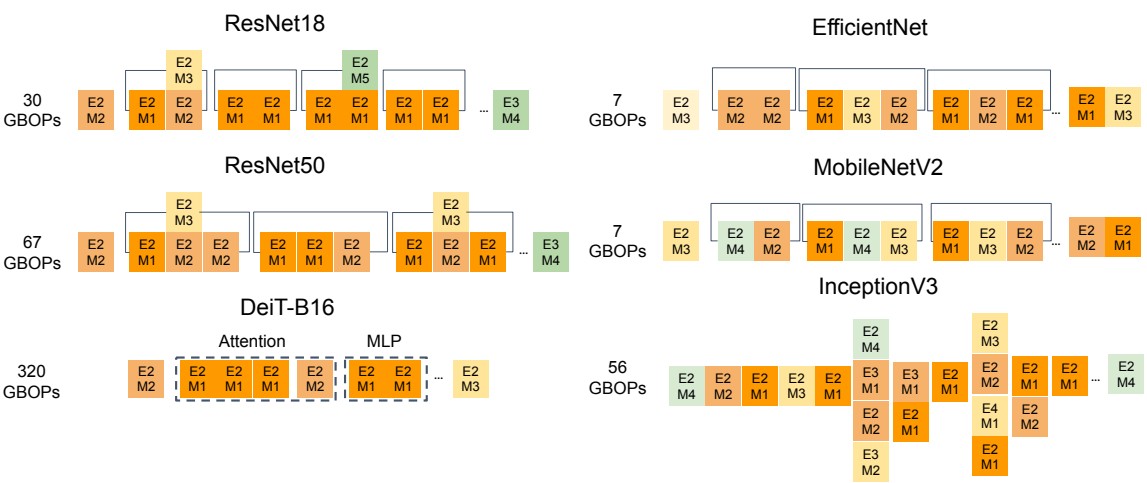

Figure 9: **Floating-Point FLIQS-L Examples**

| Start Step | Top-1 |
|---|---|
| 1000 | 75.59 |
| 2000 | 75.59 |
| 4000 | 76.11 |
| 6000 | 76.05 |
| 8000 | 76.02 |
| 10000 | 76.03 |
| 15000 | 75.94 |
| 20000 | 75.93 |
| 25000 | 75.54 |
| 30000 | 74.19 |

(a) **Start Step**

| STD Multiple | Top-1 |
|---|---|
| 1 | 63.39 |
| 2 | 67.79 |
| 3 | 68.02 |
| 4 | 67.91 |
| 5 | 67.19 |
| 6 | 67.00 |
| 7 | 66.15 |
| 8 | 64.91 |

(b) **STD Multiple**

| Profile Batches | Top-1 |
|---|---|
| 1 | 67.92 |
| 5 | 68.09 |
| 10 | 68.00 |
| 50 | 67.75 |
| 100 | 68.02 |

(c) **Profile Batches**

Figure 10: **Quantization Ablation Studies** – (a) The optimal start time for activation quantization is approximately 20% into the training process. (b) The optimal STD multiple to determine the activation clipping threshold is around 3. (c) The quantization process is relatively insensitive to the number of profiling batches.

## A.2 Two-Phase Quantization

These shared weights are quantized dynamically with a method adapted from the open-source library Accurate Quantized Training (AQT) [38], which supports both integer and emulated low-precision floating point quantization. This process can be summarized as:

$$x_q = \lfloor s \cdot \sigma(x_f; \sigma_t) \rceil \tag{10}$$
$$\sigma(x_f; \sigma_t) = \max(-\sigma_t, \min(x_f, \sigma_t)) \tag{11}$$

where $x_q$ is the quantized value, $x_f$ is the original full-precision number, $s$ is the scale factor, and $\sigma_t$ denotes the clipping threshold. In addition, $\sigma(\cdot)$ represents a clipping function, and $\lfloor \cdot \rceil$ represents a generic rounding function that pushes the value to the nearest integer or low-precision floating-point value.

The scale factor $s$ normalizes the input to the chosen maximum representable number and then rescales to the maximum quantized value. The clipping threshold and scale factor are determined by the run-time statistics of the weights and activations. Additionally, FLIQS uses a two-phase quantization approach where the weights and activations begin quantization at different training steps, as shown in Figure 2.

The two-phase quantization approach has been found empirically to improve the final accuracy of the model. In the first phase, only the weights are quantized and in the second phase, the weights and activations are quantized. The start step of the second phase has a large effect on the final accuracy. Table 10a shows the effect of sweeping the starting step for activation quantization on a ResNet50 trained to 30,200 steps. On one extreme, with the second phase starting as soon as possible, this method degenerates into a single-phase quantization method where weight and activation quantization begin immediately. On the other extreme, where the second phase begins as late as possible, it becomes a hybrid QAT-PTQ method where the weights are quantized during training and the activations are quantized after training.

Table 10a shows that accuracy peaks around 15-20% of the total training time. For this reason, FLIQS uses 7500 steps as the start step for activation quantization for ResNets and InceptionV3, which train to 30,200 steps, and 20,000 as the start step for MobileNetV2, EfficientNet, and DeiT, which train to 112,000 steps or longer.

The quantization method additionally depends on the activation clipping threshold, which is calculated as a multiple of the profiled activation standard deviations per layer. With too small a

clipping threshold, there is lower rounding error on the more numerous central values distribution, yet there is significant clipping error on the larger values. With too large a threshold, there is more rounding error on the central values and less clipping error on the larger values.

This trade-off is demonstrated empirically in Table 10b, where standard deviation multiples are swept from 1 to 8 and applied globally to a ResNet18. This table shows that the best accuracy are achieved around 3-4 times the standard deviation in ResNet18. For simplicity, we apply this multiple to all models for our experiments. Table 10c shows that the final accuracy is not sensitive to the number of profiling batches. This is likely because we use a large batch size of 2048, and since it is shuffled, it likely already provides a strong estimate of the statistics.

### A.3 Training Hyper-Parameters

The training hyper-parameters are chosen to be close to original paper hyper-parameters or recent related work. Table 4 shows the hyper-parameters used to produce Table 1 and Figure 5.

Table 4: **Training Hyper-Parameters** – Training Hyper-parameters for all quantization search table results. Same hyper-parameters are used to produce the Pareto-curve figures, although the total training time is reduced along with dependent hyper-parameters, e.g. activation quantization start step.

| Parameter | ResNets IncV3 | DeiT-B16 | MBV2 EffNet |
|---|---|---|---|
| LR Schedule | Cos | Cos | Exp |
| LR Base | 2.64 | 4e-3 | 0.256 |
| LR Warmup | 10 | 30 | 15 |
| Optimizer | SGD | AdamW | RMSProp |
| Epochs | 350 | 400 | 360 |
| Act. Quant Start | 15,000 | 15,000 | 18,000 |
| ST Multiple | 4 | 4 | 4 |

### A.4 Search Hyper-Parameters

For our search, the RL controller warmup period lasts the first 25% of the training It uses an Adam optimizer with learning rate of 4.6E-3 and momentum of .95. The loss function is a standard softmax cross entropy loss with a label smoothing coefficient set to 0.1. A cosine entropy regularization schedule is applied to all runs beginning with no regularization and ending with $\beta_H = .5$. For QNAS, during the RL controller warmup period, the branches corresponding to various kernel sizes are sampled jointly with a probability schedule. This schedule begins at 1 at the beginning of training and decreases linearly to 0 at the end of the warmup period. After the warmup period, only a single branch is sampled at a time.

### A.5 Search Space

In general, the search spaces used with FLIQS should reflect the capabilities of the target hardware. Small search spaces are useful for adapting a model to existing hardware such as the TPUv3 or NVIDIA A100. Large search spaces are useful for reconfigurable hardware such as the AMD Xilinx UltraScale+ FPGA and for co-designing models with future accelerators. The largest search space evaluated in this work includes 16 floating-point formats.

For the integer FLIQS-S search space, we include INT4, INT8, and BF16. These are the standard formats supported in modern GPU micro-architectures, such as NVIDIA Ampere. Many platforms additionally support FP16, yet this format typically performs worse than BF16 in most common use

Table 5: **Search Space**: FLIQS-S is a small search space designed to target existing hardware support, while FLIQS-L is a large search space useful for co-design with custom hardware. The floating-point FLIQS-L space demonstrates the scalability of RL-based approaches

|  | **FLIQS-S** | **FLIQS-L** |
|---|---|---|
| **Integer** | INT4, INT8, BF16 | INT4, INT5, INT6, INT7, INT8, BF16 |
| **Floating Point** | E2M1, E4M3, BF16 | E2M1, E2M2, E2M3, E2M4, E2M5, E3M1, E3M2, E3M3, E3M4, E4M1, E4M2, E4M3, E5M1, E5M2, E6M1, BF16 |

cases, so it omitted. For integer FLIQS-L, we fill in the values between INT4 and INT8 primarily considering custom hardware with integer support. For example, bit-serial deep learning engines can take advantage of this additional flexibility.

For floating-point FLIQS-S, we include three formats to be consistent with the integer search variant. BF16 is the most common half-precision format, E4M3 is the FP8 variant most useful for inference (E4M2 primarily used for gradients), and E2M1 is a custom FP4 format. For FLIQS-L, we include all the formats with total bitwidths between four and eight.

All custom formats support subnormals and do not support infinity. The bias terms are selected so the exponent range is symmetric about zero. However, this bias term is not relevant to FLIQS, since continuing from prior work [38, 6], it uses a profiled scale factor during training and search. This means that the bias term combines with the profiled scale factor and has no additional effect. Therefore, the dynamic range is controlled more by the additional scale factor than the format itself and can adequately scale to various data distributions; the format instead primarily determines the distribution of quantization points (non-linear for floating-point and linear for integer ).

### A.6 Search Performance

|  | Memory (MiB) | | | Search |
|---|---|---|---|---|
|  | Gradient | Weight | Activation | Parameters |
| FLIQS | 46.8 | 23.4 | 73.6 | 51 |
| Branched | 92.6 | 70.2 | 220.8 | 51 |

Table 6: **ResNet18 Memory** – the estimated memory breakdown for a ResNet18 model during quantization search on the FLIQS-S search space. Branched represents the class of quantization searches that create multiple branches during their search. Batch size is fixed at 32, model weights and activations are stored in half-precision, and gradients are full-precision with no gradient checkpointing. Search Parameters represents the additional parameters necessary for the search process. FLIQS and branched methods require an additional parameter for each searched layer for each searched option.

### A.7 QNAS Analysis

In general, QNAS searches tend to allocate more of their compute to architectural components, especially at high cost targets. This behavior is shown in Figure 6, where expanding quantization searches to include flexible channels and kernel size dimensions increases the accuracy of the model at similar costs. Within these architectural components, typically the channel dimension

is increased before the kernel size to reach cost targets. This could be due to the fundamental difference between kernel size and channel width; kernel size reflects the ability to aggregate information faster across spatial dimensions and channel width controls the depth of a neural network representation.

The channel dimension allocations also show an interesting trend in that lower bitwidths typically receive a larger number of channels. This is intuitive since increasing the channel width can potentially recuperate losses in representational ability from the lower bitwidth. There is a weaker trend in this direction with kernel size, where the kernel size can tend to be larger with lower bitwidths, although it is not as strong.

## A.8 Analysis Setup

For shifting error and clipping analysis, we simulate the data distributions commonly found within neural networks. For this, we use Gaussian and Laplacian distributions and inject additional outlier values. These outliers are set at 3× the maximum value in the original tensor and are injected at various rates from 1:10 to 1:10000. These outliers are especially common in activation tensors.

For the shifting error, we then sample 1000 tensors independently at random, and quantize them with two different symmetric linear quantizers that vary only in their bitwidths. We then calculate the RMS error between the two output tensors and average over all 1000 tensors. Finally, we fit the best exponential function with the form: $Ae^{(-Bx)} + C$.

Similarly, for the clipping analysis, we sample 100 tensors and calculate the quantization error between the original FP32 and quantized tensors for each percentile value. For the percentiles, we use a linear grid of 100 values from $[1, 101]$. We then plot the average MSE error over the 100 tensors and separately plot the optimal percentile. We experimented with different metrics, such as the Kullback-Liebler (KL) divergence, yet these did not lead to qualitatively different results.

## A.9 Mantissa Sweep

Table 7: **FP8 Sweep** – Sweep over possible FP8 values and evaluate Top-1 accuracy on ImageNet. All methods use an exponent bias of 11.

| Mode | ResNet18 | ResNet50 | MobileNetV2 | InceptionV3 |
|------|----------|----------|-------------|-------------|
| E1M6 | 71.72 | 77.80 | 73.20 | 76.53 |
| E2M5 | 71.70 | 77.74 | 73.14 | 76.36 |
| E3M4 | 71.69 | 77.55 | 73.17 | 76.48 |
| E4M3 | 71.69 | 77.66 | 72.65 | 76.30 |
| E5M2 | 71.59 | 76.90 | 72.07 | 76.15 |

Table 7 shows the general effects of different possible FP8 formats on ImageNet accuracy. The models are generally resilient to FP8 quantization with MobileNetV2 having the largest accuracy degradation with the E5M2 format. This is analogous to integer quantization, where typically INT8 is sufficient for most models to maintain neutral accuracy and where MobileNetV2 is more sensitive to low-bit quantization. In this table, the accuracy trends upward with more mantissa bits, and therefore not only do they determine the majority of the area in floating-point units, they increase the accuracy of the models. This leads to the classical accuracy-performance trade-off that floating-point quantization search attempts to navigate for optimal configurations. Yet for hardened accelerators, the peak throughput for different FP8 formats is the same, and therefore higher mantissa bitwidth is preferable.

## A.10 Cost Model FPGA Validation

Table 2 shows the hardware area estimates and accuracy of a set of ResNet-18 models on an AMD Xilinx UltraScale+ FPGA, implemented using Vivado HLS [39]. Since the whole model does not fit on the board, we estimate the cost with the first residual block in the model, which consists of two convolutional layers on one branch, and a pointwise convolution on the other, followed by their sum. Since all MAC operations are mapped to look-up tables (LUTs), the LUT count acts as a proxy for the area and power overhead of the model. The precision settings for FLIQS-L are taken from actual runs and represent the general bitwidth allocation to the ResNet blocks, although there may be some deviation within individual blocks.

This table shows that LUTs scale quadratically with the precision bitwidth. Since the LUTs act as a proxy for area, this verifies the core assumption of the BOPs model (Section 1) that BOPs are proportional to the model area and power on chip. This table also confirms the overhead from these searched models is indeed relatively small compared to the accuracy improvements shown in Table 1.

## A.11 Retraining vs. No Retraining

With sufficient entropy regularization, retraining the model after FLIQS is unnecessary. Table 8 shows a sweep for ResNet18 with and without retraining. With retraining, the search occurs as described in Section 3, except that the best configuration is taken and retrained from scratch for the original training length. The table shows natural variance between the retraining and no-retraining methods, but there is no noticeable advantage to retraining across model widths.

Table 8: **Retraining ResNet-18**

| | ImageNet Top1 | | | | | |
| --- | --- | --- | --- | --- | --- | --- |
| Format | 0.5× | 0.75× | 1.0× | 1.25× | 1.5× | 2.0× |
| FLIQS-S | 59.08 | 64.93 | 69.92 | 71.94 | 73.32 | 75.06 |
| + Retrain | 59.00 | 66.47 | 69.53 | 71.63 | 73.20 | 74.95 |
| FLIQS-L | 60.11 | 66.28 | 69.56 | 71.61 | 73.12 | 74.83 |
| + Retrain | 60.10 | 66.39 | 69.56 | 71.58 | 73.02 | 74.78 |

## A.12 All Models

Figure 11 plot all models with corresponding colors for methods and corresponding symbols for models. It shows that FLIQS MobileNetV2 and EfficientNet models consistently outperform other models in terms of accuracy and model cost, and BF16 models consistently perform the worst. This is expected since, as their name suggests, these models are designed specifically to be efficient and both use inverted bottleneck structures to reduce overall compute. The worst performing model overall is ResNet18, which is followed in the higher model costs by ResNet50.

## A.13 Recommendation Model

Next, we briefly explore FLIQNAS on recommendation models using the Criteo dataset [40], which is the most popular public advertisement click-through-rate (CTR) prediction benchmark. We evaluate a multi-layer perceptron (MLP) model with four hidden layers and layer factorization technique [41] similar to the method used in DCN-V2 (Deep & Cross Network) [42]. We use the AUC metric for evaluation, and list additional details about the dataset, model architecture and search space.

Figure 13 compares FLIQNAS and FLIQS with uniformly quantized models on both integer and float quantization. We focus only on FLIQS-L due to the small search space and do not include the

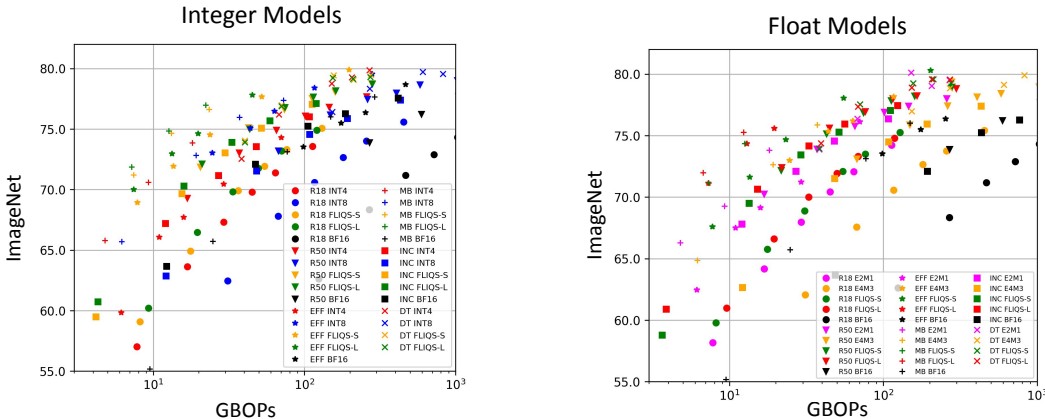

Figure 11: **Model Comparisons**: Left − *Integer All Models*. Right − *Floating-Point All Models*.

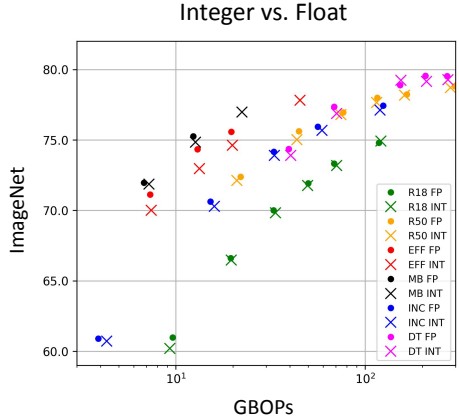

Figure 12: **Floating-Point vs. Integer FLIQS-L** − Floating-point models typically outperform their integer counter-parts.

uniformly quantized INT4 and E2M1 models since they show significant quality loss. Figure 13 shows that FLIQNAS-L performs better than FLIQS-L especially at larger MBOPs. Both of them show better quality and performance trade-offs than uniform quantization.

**Criteo**: The Criteo dataset [40] contains user logs over a period of 7 days with a total of 45M examples. Each example has 13 continuous features and 26 categorical features with a binary label indicating if an advertisement was clicked or not.

**Architecture**: The recommendation model architecture starts with an embedding layer to project the sparse categorical features into dense vectors. The embedded vectors are then concatenated with the continuous features and fed into the MLP with four hidden layers and low-rank on each layer to reduce the computational cost.

**Search Space**: For FLIQS-L, the search space uses the same configurations for integer or floating-point search on each layer. For FLIQNAS-L, besides the quantization search space, we also include $128$ and $512 \times [0.25, 0.5, 0.75, 1.0, 1.25, 1.5, 1.75, 2.0]$ for rank values and layer widths respectively on each layer.

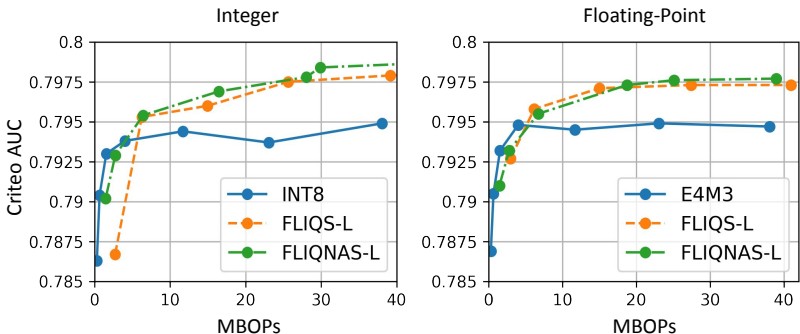

Figure 13: **Recommender FLIQNAS** – Models are trained on the Criteo dataset and evaluated by AUC (Area Under the ROC Curve) vs. millions of BOPs (MBOPs). Both FLIQNAS and FLIQS perform better than the INT8 and E4M3 baselines.

## A.14 Additional Pareto Tables

This section lists all of the raw data used to produce the Pareto curves in Figure 5.

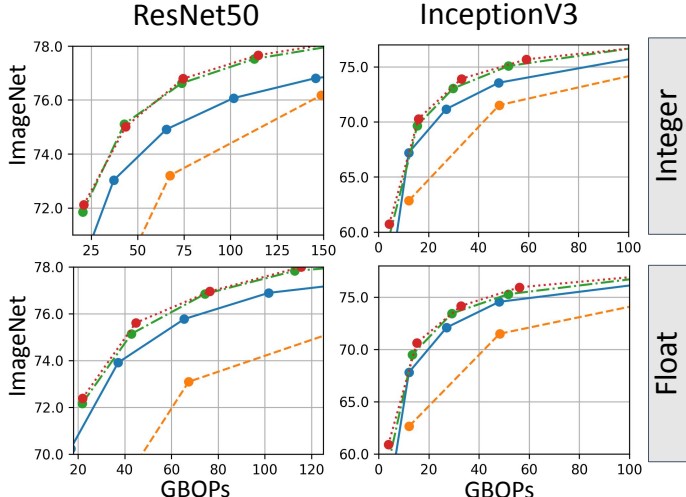

Figure 14: **Additional Pareto Curves** – Additional integer and floating-point Pareto curves that could not fit in the main paper.

| ImageNet Top1 | | | | | | |
|---|---|---|---|---|---|---|
| Format | 0.5× | 0.75× | 1.0× | 1.25× | 1.5× | 2.0× |
| BF16 | 62.63 | 68.34 | 71.17 | 72.89 | 74.31 | 75.56 |
| INT4 | 57.03 | 63.64 | 67.32 | 69.79 | 71.39 | 73.57 |
| INT8 | 62.46 | 67.80 | 70.60 | 72.65 | 74.01 | 75.58 |
| FLIQS-S | 59.08 | 64.93 | 69.92 | 71.94 | 73.32 | 75.06 |
| FLIQS-L | 60.11 | 66.28 | 69.56 | 71.61 | 73.12 | 74.83 |
| + ($\beta_H^{cos}$) | 60.21 | 66.47 | 69.83 | 71.76 | 73.19 | 74.91 |

| GBOPs | | | | | | |
|---|---|---|---|---|---|---|
| Format | 0.5× | 0.75× | 1.0× | 1.25× | 1.5× | 2.0× |
| BF16 | 124.5 | 268.8 | 467.7 | 721.3 | 1030 | 1810 |
| INT4 | 7.78 | 16.80 | 29.23 | 45.08 | 64.35 | 113.1 |
| INT8 | 31.13 | 67.19 | 116.9 | 180.3 | 257.4 | 452.5 |
| FLIQS-S | 8.18 | 17.68 | 36.46 | 54.60 | 76.35 | 130.7 |
| FLIQS-L | 9.04 | 19.09 | 32.33 | 49.22 | 69.20 | 120.2 |
| + ($\beta_H^{cos}$) | 9.30 | 19.58 | 33.54 | 49.57 | 70.54 | 120.8 |

Figure 15: **Integer ResNet-18**

| ImageNet Top1 | | | | | | |
|---|---|---|---|---|---|---|
| Format | 0.5× | 0.75× | 1.0× | 1.25× | 1.5× | 2.0× |
| INT4 | 69.27 | 73.03 | 74.91 | 76.07 | 76.81 | 77.68 |
| INT8 | 73.20 | 76.17 | 77.47 | 77.98 | 78.66 | 79.00 |
| FLIQS-S | 71.85 | 75.11 | 76.62 | 77.52 | 78.06 | 78.76 |
| FLIQS-L | 71.56 | 74.67 | 76.52 | 77.37 | 78.02 | 78.73 |
| + ($\beta_H^{cos}$) | 72.12 | 75.01 | 76.79 | 77.66 | 78.17 | 78.72 |
| BF16 | 73.87 | 76.22 | 77.68 | 78.45 | 78.82 | 79.14 |

| GBOPs | | | | | | |
|---|---|---|---|---|---|---|
| Format | 0.5× | 0.75× | 1.0× | 1.25× | 1.5× | 2.0× |
| INT4 | 16.84 | 37.16 | 65.43 | 101.6 | 145.8 | 257.9 |
| INT8 | 67.35 | 148.7 | 261.7 | 406.5 | 583.1 | 1031 |
| FLIQS-S | 20.49 | 42.87 | 73.66 | 112.7 | 160 | 279.3 |
| FLIQS-L | 20.51 | 41.13 | 71.57 | 112.9 | 156.4 | 273.5 |
| + ($\beta_H^{cos}$) | 21.03 | 43.55 | 74.49 | 114.8 | 161.5 | 282 |
| BF16 | 269.4 | 594.6 | 1047 | 1626 | 2332 | 4126 |

Figure 16: **Integer ResNet-50**

| ImageNet Top1 | | | | | |
|---|---|---|---|---|---|
| Format | 0.25× | 0.5× | 1.0× | 1.4× | 2.0× |
| BF16 | 55.18 | 65.72 | 73.13 | 76.00 | 77.64 |
| INT4 | 40.62 | 54.11 | 65.80 | 70.60 | 73.85 |
| INT8 | 55.09 | 65.70 | 72.83 | 75.95 | 77.37 |
| FLIQS-S | 50.78 | 63.03 | 71.21 | 74.64 | 76.61 |
| FLIQS-L | 52.38 | 63.15 | 71.73 | 74.99 | 77.01 |
| + $(\beta_H^{\cos})$ | 52.11 | 63.35 | 71.87 | 74.83 | 76.98 |

| GBOPs | | | | | |
|---|---|---|---|---|---|
| Format | 0.25× | 0.5× | 1.0× | 1.4× | 2.0× |
| BF16 | 9.52 | 24.87 | 77.00 | 149.0 | 291.2 |
| INT4 | 0.595 | 1.55 | 4.81 | 9.31 | 18.20 |
| INT8 | 2.38 | 6.21 | 19.25 | 37.20 | 72.80 |
| FLIQS-S | 1.16 | 2.90 | 7.42 | 13.21 | 23.51 |
| FLIQS-L | 1.06 | 2.38 | 7.06 | 12.70 | 22.26 |
| + $(\beta_H^{\cos})$ | 1.02 | 2.42 | 7.21 | 12.69 | 22.31 |

Figure 17: **Integer MobileNetV2**

| ImageNet Top1 | | | | | |
|---|---|---|---|---|---|
| Format | 0.25× | 0.5× | 0.75× | 1.0× | 1.5× |
| BF16 | 63.65 | 72.10 | 75.24 | 76.26 | 77.55 |
| INT4 | 53.20 | 67.20 | 71.16 | 73.55 | 76.00 |
| INT8 | 62.86 | 71.52 | 74.56 | 75.87 | 77.38 |
| FLIQS-S | 59.49 | 69.66 | 73.04 | 75.07 | 77.05 |
| + $(\beta_H^{\cos})$ | 60.72 | 70.28 | 73.91 | 75.67 | 77.12 |

| GBOPs | | | | | |
|---|---|---|---|---|---|
| Format | 0.25× | 0.5× | 0.75× | 1.0× | 1.5× |
| BF16 | 48.69 | 193.1 | 433.3 | 769.2 | 1728 |
| INT4 | 3.04 | 12.07 | 27.08 | 48.08 | 108.0 |
| INT8 | 12.17 | 48.28 | 108.3 | 192.3 | 432.0 |
| FLIQS-S | 4.18 | 15.53 | 29.88 | 52.02 | 112.5 |
| + $(\beta_H^{\cos})$ | 4.31 | 15.99 | 33.17 | 59.16 | 119.9 |

Figure 18: **Integer InceptionV3**

| ImageNet Top1 | | | | | |
|---|---|---|---|---|---|
| Format | B0 | B1 | B2 | B3 | B4 |
| BF16 | 73.53 | 75.50 | 76.36 | 78.68 | 80.35 |
| INT4 | 59.83 | 66.08 | 67.71 | 70.46 | 74.29 |
| INT8 | 73.04 | 75.08 | 76.48 | 78.39 | 79.55 |
| FLIQS-S | 68.94 | 71.92 | 74.53 | 77.67 | 79.89 |
| FLIQS-L | 70.51 | 73.23 | 75.41 | 77.96 | 80.03 |
| + $(\beta_H^{\cos})$ | 70.01 | 72.96 | 74.62 | 77.81 | 79.92 |

| GBOPs | | | | | |
|---|---|---|---|---|---|
| Format | B0 | B1 | B2 | B3 | B4 |
| BF16 | 98.61 | 175.5 | 254.0 | 467.3 | 1124 |
| INT4 | 6.16 | 10.97 | 15.88 | 29.21 | 70.25 |
| INT8 | 24.65 | 43.89 | 63.50 | 116.8 | 281.0 |
| FLIQS-S | 7.86 | 13.62 | 23.81 | 52.30 | 198.0 |
| FLIQS-L | 7.40 | 13.21 | 21.62 | 49.38 | 187.0 |
| + $(\beta_H^{\cos})$ | 7.42 | 13.32 | 19.85 | 45.26 | 187.1 |

Figure 19: **Integer EfficientNet**

| ImageNet Top1 | | | | | | |
|---|---|---|---|---|---|---|
| Format | 0.25× | 0.375× | 0.5× | 0.75× | 0.875× | 1.0× |
| INT4 | 66.51 | 72.53 | 76.19 | 78.75 | 79.26 | 79.84 |
| INT8 | 70.77 | 76.41 | 78.33 | 79.71 | 79.55 | 79.49 |
| FLIQS-S | 66.36 | 74.05 | 76.96 | 79.44 | 79.05 | 79.47 |
| FLIQS-L | 67.04 | 73.93 | 77.10 | 79.27 | 79.27 | 79.35 |
| + $(\beta_H^{\cos})$ | 67.78 | 73.90 | 76.88 | 79.23 | 79.16 | 79.28 |

| GBOPs | | | | | | |
|---|---|---|---|---|---|---|
| Format | 0.25× | 0.375× | 0.5× | 0.75× | 0.875× | 1.0× |
| INT4 | 20.29 | 38.42 | 67.96 | 152.1 | 206.8 | 269.8 |
| INT8 | 80.94 | 153.5 | 271.5 | 608.1 | 826.6 | 1079 |
| FLIQS-S | 20.31 | 40.52 | 70.74 | 156.3 | 211.7 | 275.4 |
| FLIQS-L | 21.08 | 39.55 | 69.12 | 153.9 | 208.8 | 272.0 |
| + $(\beta_H^{\cos})$ | 21.47 | 40.45 | 70.57 | 154.5 | 210.4 | 273.2 |

Figure 20: **Integer DeiT-B16**

| ImageNet Top1 | | | | | | |
|---|---|---|---|---|---|---|
| **Format** | **0.5×** | **0.75×** | **1.0×** | **1.25×** | **1.5×** | **2.0×** |
| BF16 | 62.63 | 68.34 | 71.17 | 72.89 | 74.31 | 75.56 |
| E2M1 | 58.17 | 64.18 | 67.96 | 70.43 | 72.08 | 74.22 |
| E4M3 | 62.06 | 67.57 | 70.56 | 72.66 | 73.75 | 75.43 |
| FLIQS-S | 59.80 | 65.77 | 68.89 | 72.10 | 73.50 | 75.26 |
| + ($\beta_H^{cos}$) | 60.99 | 66.61 | 70.01 | 71.92 | 73.32 | 74.80 |

| GBOPs | | | | | | |
|---|---|---|---|---|---|---|
| **Format** | **0.5×** | **0.75×** | **1.0×** | **1.25×** | **1.5×** | **2.0×** |
| BF16 | 124.5 | 268.8 | 467.7 | 721.3 | 1030 | 1810 |
| E2M1 | 7.78 | 16.80 | 29.23 | 45.08 | 64.35 | 113.1 |
| E4M3 | 31.13 | 67.19 | 116.9 | 180.3 | 257.4 | 452.5 |
| FLIQS-S | 8.18 | 17.68 | 30.80 | 54.60 | 76.35 | 128.2 |
| + ($\beta_H^{cos}$) | 9.60 | 19.48 | 32.78 | 50.01 | 68.43 | 118.2 |

Figure 21: **Floating-Point ResNet-18**

| ImageNet Top1 | | | | | | |
|---|---|---|---|---|---|---|
| **Format** | **0.5×** | **0.75×** | **1.0×** | **1.25×** | **1.5×** | **2.0×** |
| BF16 | 73.87 | 76.22 | 77.68 | 78.45 | 78.82 | 79.14 |
| E2M1 | 70.24 | 73.91 | 75.77 | 76.89 | 77.40 | 78.01 |
| E4M3 | 73.09 | 75.86 | 77.42 | 78.13 | 78.42 | 78.97 |
| FLIQS-S | 72.16 | 75.14 | 76.84 | 77.83 | 78.22 | 78.94 |
| + ($\beta_H^{cos}$) | 72.39 | 75.61 | 76.95 | 78.00 | 78.24 | 78.81 |

| GBOPs | | | | | | |
|---|---|---|---|---|---|---|
| **Format** | **0.5×** | **0.75×** | **1.0×** | **1.25×** | **1.5×** | **2.0×** |
| BF16 | 269.4 | 594.6 | 1047 | 1626 | 2332 | 4126 |
| E2M1 | 16.84 | 37.16 | 65.43 | 101.6 | 145.8 | 257.9 |
| E4M3 | 67.35 | 148.7 | 261.7 | 406.5 | 583.1 | 1031 |
| FLIQS-S | 21.72 | 42.87 | 74.28 | 112.7 | 160.0 | 279.2 |
| + ($\beta_H^{cos}$) | 21.93 | 44.70 | 76.43 | 115.6 | 165.6 | 297.9 |

Figure 22: **Floating-Point ResNet-50**

| ImageNet Top1 | | | | |
|---|---|---|---|---|
| **Format** | **0.25×** | **0.5×** | **1.0×** | **1.4×** |
| BF16 | 55.18 | 65.72 | 73.13 | 76.00 |
| E2M1 | — | 52.32 | 66.29 | 69.26 |
| E4M3 | 53.98 | 64.85 | 72.63 | 75.86 |
| FLIQS-S | 50.76 | 62.58 | 71.14 | 74.34 |
| + ($\beta_H^{cos}$) | 51.11 | 63.65 | 71.97 | 75.26 |

| GBOPs | | | | |
|---|---|---|---|---|
| **Format** | **0.25×** | **0.5×** | **1.0×** | **1.4×** |
| BF16 | 9.52 | 24.87 | 77.00 | 149.0 |
| E2M1 | 0.595 | 1.55 | 4.81 | 9.31 |
| E4M3 | 2.38 | 6.21 | 19.25 | 37.20 |
| FLIQS-S | 1.22 | 2.69 | 7.35 | 12.6 |
| + ($\beta_H^{cos}$) | 0.95 | 2.36 | 6.77 | 12.37 |

Figure 23: **Floating-Point MobileNetV2**

| ImageNet Top1 | | | | | |
|---|---|---|---|---|---|
| **Format** | **0.25×** | **0.5×** | **0.75×** | **1.0×** | **1.5×** |
| BF16 | 63.65 | 72.10 | 75.24 | 76.26 | 77.55 |
| E2M1 | 54.14 | 67.80 | 72.09 | 74.55 | 76.35 |
| E4M3 | 62.65 | 71.50 | 74.49 | 75.94 | 77.39 |
| FLIQS-S | 58.78 | 69.50 | 73.43 | 75.28 | 77.03 |
| + ($\beta_H^{cos}$) | 60.90 | 70.63 | 74.16 | 75.94 | 77.43 |

| GBOPs | | | | | |
|---|---|---|---|---|---|
| **Format** | **0.25×** | **0.5×** | **0.75×** | **1.0×** | **1.5×** |
| BF16 | 48.69 | 193.1 | 433.3 | 769.2 | 1728 |
| E2M1 | 3.04 | 12.07 | 27.08 | 48.08 | 108.0 |
| E4M3 | 12.17 | 48.28 | 108.3 | 192.3 | 432.0 |
| FLIQS-S | 3.66 | 13.46 | 29.16 | 51.65 | 111.4 |
| + ($\beta_H^{cos}$) | 3.89 | 15.23 | 32.86 | 56.15 | 124.2 |

Figure 24: **Floating-Point InceptionV3**

| ImageNet Top1 | | | | | |
|---|---|---|---|---|---|
| **Format** | **B0** | **B1** | **B2** | **B3** | **B4** |
| BF16 | 73.53 | 75.50 | 76.36 | — | — |
| E2M1 | 62.45 | 67.49 | 69.14 | 71.22 | 76.12 |
| E4M3 | 72.99 | 75.36 | 76.20 | 78.17 | 79.52 |
| FLIQS-S | 67.60 | 71.63 | 74.67 | 78.05 | 80.30 |
| + ($\beta_H^{cos}$) | 71.13 | 74.34 | 75.58 | 78.03 | 80.29 |

| GBOPs | | | | | |
|---|---|---|---|---|---|
| **Format** | **B0** | **B1** | **B2** | **B3** | **B4** |
| BF16 | 98.61 | 175.5 | 254.0 | — | — |
| E2M1 | 6.16 | 10.97 | 15.88 | 29.21 | 70.25 |
| E4M3 | 24.65 | 43.89 | 63.50 | 116.8 | 281.0 |
| FLIQS-S | 7.77 | 13.58 | 23.26 | 55.15 | 203.3 |
| + ($\beta_H^{cos}$) | 7.30 | 13.00 | 19.61 | 36.6 | 212.9 |

Figure 25: **Floating-Point EfficientNet**

| ImageNet Top1 | | | | | | |
|---|---|---|---|---|---|---|
| **Format** | **0.25×** | **0.375×** | **0.5×** | **0.75×** | **0.875×** | **1.0×** |
| E2M1 | 66.63 | 73.88 | 76.35 | 80.10 | 79.04 | 79.49 |
| E4M3 | 71.19 | 76.96 | 78.85 | 79.13 | 79.90 | 79.17 |
| FLIQS-S | 67.27 | 73.95 | 77.52 | 79.24 | 79.56 | 79.27 |
| FLIQS-L | 68.25 | 74.36 | 77.35 | 78.89 | 79.56 | 79.54 |

| GBOP | | | | | | |
|---|---|---|---|---|---|---|
| **Format** | **0.25×** | **0.375×** | **0.5×** | **0.75×** | **0.875×** | **1.0×** |
| E2M1 | 20.29 | 38.42 | 67.96 | 152.1 | 206.8 | 269.8 |
| E4M3 | 80.94 | 153.5 | 271.5 | 608.1 | 826.6 | 1079 |
| FLIQS-S | 20.3 | 38.42 | 70.74 | 156.3 | 211.7 | 275.4 |
| FLIQS-L | 21.08 | 39.40 | 68.49 | 152.9 | 207.7 | 270.8 |

Figure 26: **Floating-Point DeiT-B16**

