# OpenReview forum: "FLIQS: One-Shot Mixed-Precision Floating-Point and Integer Quantization Search"
_automl.cc/AutoML/2024/Conference — AutoML 2024_

### Official Review · Reviewer_Yor1 · 2024-03-28

**Potential Impact On The Field Of Automl Rating:** 3
**Technical Quality And Correctness:** 1. The authors applied existing appro…
**Technical Quality And Correctness Rating:** 3
**Clarity:** n/a
**Clarity Rating:** 3
**Actions Required To Increase Overall Recommendation:** n/a

**Summary Of Contributions:**

This paper proposed to search for mixed-precision and quantization networks based on a one-shot and RL-based search algorithm.

**Overall Review:**

This paper is well-motivated and well-organized.  Although its algorithm novelty is not very high, I'd like to see the authors apply NAS method to new applications. Experiments are solid. I believe this paper should be accepted.

**Potential Impact On The Field Of Automl:**

It is well-motivated to search for mixed-precision and integer neural networks as more hardware emerges. We should encourage the authors to push the boundaries of NAS and apply NAS to fresh applications.

**Review Confidence:**

2

**Review Rating:**

7

**Review Summary:**

see Overall Review

---

### Official Review · Reviewer_4eV1 · 2024-03-28

**Potential Impact On The Field Of Automl Rating:** 3
**Technical Quality And Correctness Rating:** 3
**Clarity Rating:** 3
**Actions Required To Increase Overall Recommendation:** Please consider my previous comments …

**Summary Of Contributions:**

This paper introduces FLIQS, a technique for automated mixed-precision quantization, which involves both floating-point and integer formats. FLIQS employs a layer-wise approach and utilizes a one-shot reinforcement learning search method to optimize both the model architecture and the quantization formats simultaneously. Compared to traditional post-training quantization methods, FLIQS demonstrates higher accuracy performance.

**Clarity:**

In general, the paper is well-written and easy-to-follow. I have a few minor suggestions that could enhance the clarity of the paper:

- Introduction-> “3. Perform the largest comparison of integer and floating-point mixed precision networks”
    - Do authors mean they have compared integer networks with floating-point mixed-precision networks?

- I'm not quite sure what the authors intend to convey with Figure 1.a. Could you clarify what the authors expect reviewers to interpret from this specific figure?
- Related Work-> The literature review could be enhanced by including studies on quantization-aware NAS, which have not been extensively covered. Incorporating the following related articles could enrich the comprehensiveness of the review. Plus, it would be beneficial to explain the distinctions between FLIQS and these existing approaches.
    - Loni, Mohammad, et al. "Tas: ternarized neural architecture search for resource-constrained edge devices." 2022 Design, Automation & Test in Europe Conference & Exhibition (DATE). IEEE, 2022.
    - Dong, Peijie, et al. "Emq: Evolving training-free proxies for automated mixed precision quantization." Proceedings of the IEEE/CVF International Conference on Computer Vision. 2023.
    - Koryakovskiy, Ivan, et al. "One-shot model for mixed-precision quantization." Proceedings of the IEEE/CVF Conference on Computer Vision and Pattern Recognition. 2023.
    - Sridhar, Sharath Nittur, et al. "SimQ-NAS: Simultaneous Quantization Policy and Neural Architecture Search." arXiv preprint arXiv:2312.13301 (2023).
    - Huang, Wei, et al. "OHQ: On-chip Hardware-aware Quantization." arXiv preprint arXiv:2309.01945 (2023).
    - Ma, Yuexiao, et al. "Ompq: Orthogonal mixed precision quantization." Proceedings of the AAAI conference on artificial intelligence. Vol. 37. No. 7. 2023.
    - Bablani, Deepika, et al. "Efficient and effective methods for mixed precision neural network quantization for faster, energy-efficient inference." arXiv preprint arXiv:2301.13330 (2023).

- Table 2-> Does LUT stand for Look-Up-Table in FPGAs?  Then what is the specification of the FPGA device?
- I cannot find any analysis on the ablation results reported in Figure 10.

**Overall Review:**

# Strengths:
- Paper is well-written and addresses important research questions.
- The proposed method does not require re-training.

# Weaknesses:
- Novelty of the paper is under question. I would like to hear from authors in which way their proposed method differs from SoTA papers (some are listed in the Clarity section).
- Compared to related studies, results are limited to CNN and ViT models. I would like to see if (and how) the proposed method could be extended for language models.
    - Sridhar, Sharath Nittur, et al. "SimQ-NAS: Simultaneous Quantization Policy and Neural Architecture Search." arXiv preprint arXiv:2312.13301 (2023).
- Table 1: Compare the results of FLIQS with:
    - Dong, Peijie, et al. "Emq: Evolving training-free proxies for automated mixed precision quantization." Proceedings of the IEEE/CVF International Conference on Computer Vision. 2023.
    - Huang, Wei, et al. "OHQ: On-chip Hardware-aware Quantization." arXiv preprint arXiv:2309.01945 (2023).
    - Bablani, Deepika, et al. "Efficient and effective methods for mixed precision neural network quantization for faster, energy-efficient inference." arXiv preprint arXiv:2301.13330 (2023).
- Reproducibility of results is under question.
- Comparison baselines are relatively old.

**Potential Impact On The Field Of Automl:**

- Green AutoML: this paper aims to minimize the expenses associated with the model tuning.

- This paper focuses on sustainability by enhancing the performance of quantized models, thereby encouraging greater usage of QNNs by engineers. Essentially, this research contributes to the increased energy efficiency of DNNs.

**Reproducibility:**

The code of the paper is not available. Thus, evaluating the reproducibility of results is impossible.

**Review Confidence:**

4

**Review Rating:**

7

**Review Summary:**

The proposal of a one-shot NAS method for designing a mixed-precision CNN/ViT is interesting, and I found the paper engaging to read. However, there are several areas where the paper could be strengthened.
- Firstly, extending the results to include language models like BERT would enhance its applicability, although I understand this may not be feasible for this submission.
- Secondly, evaluating latency results on other platforms such as CPUs or FPGAs would provide a more comprehensive understanding of the method's performance across different hardware.
- Lastly, comparing the proposed method with state-of-the-art techniques published in 2023 and 2024 would help its effectiveness in the field.

**Technical Quality And Correctness:**

Overall, the approach and experimentation are technically correct and sound. I do have a question regarding the proposed method. Based on Figure 1, it seems that FLIQS is capable of designing architectures for different hardware (HW) devices, like FPGAs. My question is whether FLIQS requires a multi-objective search method to take into account the resource constraints of the target HW device?

---

### Official Review · Reviewer_4KXy · 2024-03-28

**Potential Impact On The Field Of Automl Rating:** 3
**Technical Quality And Correctness Rating:** 3
**Clarity Rating:** 3

**Summary Of Contributions:**

The paper presents FLIQS which is the first one-shot mixed-precision quantization search to eliminate the need for retraining in both integer and low-recision floating point models. The method is evaluated fairly exhaustively across different model types (transformers and cnns) and 2 different search space types and achieves quantized models with lower latencies without additional retraining costs.

**Actions Required To Increase Overall Recommendation:**

Check limitations and questions in "Overall Review" and "Technical Quality And Correctness" sections. If authors respond adequately to my concerns I am willing to increase my score

**Clarity:**

The paper is written clearly in most parts and the details regarding search spaces, architectures obtained, hyperparameters are provided adequately. Refer to "Technical Quality And Correctness" for some questions regarding the shared model training pipeline. I encourage the authors to include more algorithmic details of the search procedure to the paper.

**Overall Review:**

Positives:
1. Strong motivation of the problem
2. Presentation is clear in most parts
3. Evaluation across architecture types is fairly exhaustive
5. Strong empirical results

Negatives
1. Some parts of the search pipeline are unclear (check Technical Quality And Correctness)
2. 2-stage one-shot methods and RL-based search algorithms are quite well studied in literature and could be applied for quantization search too. Though the paper is the first to propose a one-shot model (where in archs don't require retraining), I think the approach should compare to baselines like random-search, evolutionary-search on the shared model too.

**Potential Impact On The Field Of Automl:**

I think this paper does have a moderate impact on the field of AutoML. Specifically one-shot mixed-precision quantization search is quite novel in my opinion and opens up possibilities for joint architecture + quantization search. Furthermore, since the architectures don't require further retraining this approach is also very much aligned with "Green AutoML" and need to reduced carbon consumptions in deep learning models (due to the efficiency of the method and the lower energy consumption of architectures discovered)

**Reproducibility:**

The paper has code available and reports the hyperparameters and training details. Based on this I would rate the reproducibility of the results in the paper to be high.

**Review Confidence:**

4

**Review Rating:**

7

**Review Summary:**

This paper makes an important contribution in studying neural architecture search methods in a very large space of quantizations. The method is studied across architecture types, different quantization search spaces, and metrics (GBOPs,latency speedups). Moreover, since the architectures don't require further retraining this approach is also very much aligned with "Green AutoML" and need to reduced carbon consumptions in deep learning models (due to the efficiency of the method and the lower energy consumption of architectures discovered)

**Technical Quality And Correctness:**

- The paper is written clearly in terms of technical details and the experimental evaluation is exhaustive
- The method is explained in a clear and concise manner

Certain parts of the paper are however unclear to me:
1. The search procedure has 2 phases (a) Train the shared model with random sampling of quantizations from the search space. In this phase only the weights are quantized (b) Search+Train: The RL controller starts guiding the search and the activations are also quantized in this stage. Is my overall understanding correct?
2. I am confused by the training procedure of the shared model. Is there a single model (shared weights) which is, depending on the architecture choice quantized differently every time (similar to what OFA[1] does for weights)? Since the weights are shared does this cause conflicting gradients during training (if different quantizations are sampled at every step)?

Evaluation
1. There is plethora of research on using a one-shot supernet to perform search of architectures eg: [1,2,3] and can be also extended to search for quantization similar to the proposed metho. Hence such 2-stage methods (supernet training+black-box search) are natural candidates for baselines in this paper, however, I don't see comparisons to such approaches in the paper.
2. There are two possible ways to achieve latency reductions in this case (a) structural pruning (using NAS) (b) quantization. How do these approaches compare against each other? Is it more beneficial to search for the right quantization types or the right architectures to get the most latency gains?

[1] Cai, H., Gan, C., Wang, T., Zhang, Z. and Han, S., 2019. Once-for-all: Train one network and specialize it for efficient deployment. arXiv preprint arXiv:1908.09791.

[2] Chen, M., Peng, H., Fu, J. and Ling, H., 2021. Autoformer: Searching transformers for visual recognition. In Proceedings of the IEEE/CVF international conference on computer vision (pp. 12270-12280).

[3] Wang, H., Wu, Z., Liu, Z., Cai, H., Zhu, L., Gan, C. and Han, S., 2020. Hat: Hardware-aware transformers for efficient natural language processing. arXiv preprint arXiv:2005.14187.

---

### Official Review · Reviewer_rnwf · 2024-03-28

**Potential Impact On The Field Of Automl Rating:** 4
**Technical Quality And Correctness Rating:** 4
**Clarity Rating:** 4

**Summary Of Contributions:**

The authors consider quantization, a common method to reduce costs and runtime for model inference. The paper a/ introduces the first one-shot quantization search that doesn't require retraining by adding a cosine entropy regularization schedule, b/ compares both integer and low-precision floating-point quantization search across network types, b/ performs the largest comparison of integer and floating-point mixed-precision networks, and d/ studies quantization and NAS on low-precision floating-point networks and establishes compute resource recommendations.

**Actions Required To Increase Overall Recommendation:**

Mention novelty with respect to this newer paper (https://arxiv.org/pdf/2401.01543.pdf) which is within the same field, although FLIQS provides stronger results for a more impactful problem statement.

**Clarity:**

The paper establishes a clear problem statement and logically dives deep into methods and experiment setting. It then provides a comprehensive comparison/analysis of results in order to establish confidence in automated quantization search and establish SOTA performance in Resnet/MobileNet settings.

**Overall Review:**

+ Introduces an automated quantization search for both integer and floating point models
+ Introduces a reinforcement-learning based controller
+ Establishes impactful SOTA performance on model architectures with quantization
+ Does not require re-training, resulting in lower costs / improved productivity
- Does not comment on outcomes/findings from ablation study in body of paper

**Potential Impact On The Field Of Automl:**

The paper would be impactful to the field of quantization and a strong example of automated search leading to state-of-the art performance. Previous methods either compromises on accuracy (post-training quantization), or is unscalable from a memory standpoint (differentiable search). The paper introduces a one-shot mixed-precision quantization search that does not require retraining. A similar paper has since come out (https://arxiv.org/pdf/2401.01543.pdf) but is more narrow, does not deal with integer and floating point, and doesn't establish SOTA results on ResNet/Mobilenet, like in this paper.

**Review Confidence:**

4

**Review Rating:**

9

**Review Summary:**

The paper presents an impactful contribution to the field of neural network contribution, and is the first to do so for both integer and floating point; specifically, it outlines the first one-shot quantization search that doesn't require retraining via an RL-based controller, which would greatly lower costs / challenges with previous quantization methods. The paper provides a strong experiment setting and detailed comparisons to confidently establish state-of-the-art performance in quantized ResNet/MobileNet, and draws meaningful conclusions on trends encountered during experimentation.

**Technical Quality And Correctness:**

The paper establishes valid baselines and in-depth comparisons, and supplements with clear metrics showing improved performance from their method. It also considers the drawbacks of their approach (search interfering with model training and describe the corresponding change in weight/activations by this quantization search, noise in reward signal evaluation happens across batches of different quality due to the model configuration being considered). The authors dive deep into the different components (cost/reward function, reinforcement-learning controller, etc.). One consideration is lack of mention of high-level conclusions from ablation study in the body of the paper, rather than appendix, but this is minor.

---

### Meta-Review · Area_Chair_rQwu · 2024-04-22

**Paper Recommendation:** Accept
**Confidence:** 4

**Metareview:**

The paper describes a new neural architecture search (NAS) method for mixed floating point and integer precision to quantize network.
All reviewers are positive about the paper. Reviewer rnwf and 4KXy also highlight the strong experimental results. Overall, I think this paper could have a positive impact on NAS to stir the development of automated methods for quantization, which arguable becomes increasingly more important with ever larger networks.

---

### Decision · Program_Chairs · 2024-04-29

**Decision:**

Accept

**Comment:**

Thank you for submitting your paper. We are happy to tell you that we accept your paper to the main track. See you in Paris.